# What matters for international consumers' choice preferences for smartphones: Evidence from a cross-border ecommerce platform

**Karamoko N'da** [1]*, **Jiaoju Ge** [1], **Steven Ji-Fan Ren** [1], **Jia Wang** [2]

**1** School of Economics and Management, Harbin Institute of Technology (Shenzhen), Shenzhen, China,
**2** School of E-Commerce and Logistics, Suzhou Institute of Trade & Commerce, Suzhou, China

These authors contributed equally to this work.
* andabuy@yahoo.com

**Data Availability Statement:** All relevant data are within the paper and its Supporting Information files.

## Abstract

Despite the growing impact of smartphone use on countries' economies, the literature has rarely investigated the link between economic context and smartphone purchase trends. Based on 20,556 smartphones sold from a Cross-Border E-Commerce (CBEC) platform, the study reveals that relationships between GDP per capita and Smartphone Choice Preferences (SCP) as well as Purchase Quantities (PUR) are direct and partially mediated by Price (PRI), Read-Only Memory (ROM), and Random-Access Memory (RAM). That means that the economic context highlighted by the GDP plays a substantial role in smartphone choices and purchases. The study suggests that e-sellers and smartphone brands should adapt their marketing and manufacturing strategies to the countries' economic contexts to leverage the fearless competition in the smartphone industry.

## 1. Introduction

With digital development and its impact on society, smartphones have become indispensable tools in people's everyday lives [1]. Today, smartphones and their apps are almost indispensable for efficiently accomplishing tasks such as managing messages, emails, images, videos, geolocation, or performing online purchases [1]. Their impact on society has been even more significant during the COVID-19 outbreak, when smartphones served as open windows to the globe, allowing individuals to communicate across distances and keep informed [2]. In this context, research shows that smartphone demand in the global market is increasing steadily every year [3]. It is expected that 5.9 billion people worldwide will own smartphones in 2025 [4]. Therefore, smartphone subjects have become topics of practical and theoretical importance [1, 5]. Nevertheless, the number of studies on smartphone purchase-related subjects is still tiny compared to their importance in consumers' daily lives in this current digital era [6]. As a result, researchers have called for more research on smartphone purchases and preferences [1] to better comprehend and address consumers' needs [1, 6, 7].

**Funding:** This study, is financially sustained by the National Natural Science Foundation of China (Project No. 71402039 and Project No. 71831005), Shenzhen Peacock Program (Project No. KOCX2015032715503970). The funders had no role in study design, data collection and analysis, decision to publish, or preparation of the manuscript.

**Competing interests:** The authors have declared that no competing interests exist.

The event of smartphones and their technological evolution have undoubtedly influenced the entire society [8], including the economic context [9–11], purchase tendencies, and shopping behaviors of consumers [6]. That being so, a question remains: given smartphones' growing impact on society's economic setting [12, 13], isn't it reasonable to know if society's economic context could also influence smartphone purchase tendencies and choice preferences? An answer to such a question could be an opportunity to better understand smartphone purchase tendencies, behaviors, preferences, and decision mechanisms in different societies and economic settings [6]. However, until now, literature has rarely explored those links. Therefore, it is imperative to investigate those linkages [6], as a lack of empirical research exists on those relationships [1].

Research on smartphone purchases and preferences has been approached mainly from purchases behaviors, attitudes, and pricing angles [14–20]. Nevertheless, one notes relatively scarce research on purchase and choice decisions from a macroeconomic and cross-border perspective [21]. Not many studies have combined the macroeconomic setting, cross-border e-commerce (CBEC), and smartphone choice preferences and purchases [22]. Therefore, there is a research gap regarding the choice preferences of smartphones across different macroeconomic contexts via CBEC [22]. Most past studies were carried out through offline purchases in national settings [19]. However, in the absence of certain types of smartphones on national markets, possible buyers may only rely on cross-border markets.

Research shows that smartphone purchases are experiencing unprecedented growth in CBEC marketplaces [23]. However, despite those unprecedented purchases through CBEC marketplaces and their spread in various macroeconomic settings worldwide, little research has been interested in them [23]. Therefore, knowledge of Smartphone Choice Preferences (SCP) and Purchase Quantities (PUR) via cross-border marketplaces in consideration of macroeconomic settings is needed. Only a few researches have emphasized the significance that the macroeconomic setting could have on smartphone purchase preferences [24]. Most of those studies have been carried out based on GDP per capita or income level [22]. However, according to the macroeconomic background, the results were inconclusive or opposed to each other. For instance, the impact of GDP per capita on android smartphone purchases was moderately positive in underdeveloped countries [22]. At the same time, the relationship was relatively negative in developed countries and insignificant globally [22]. A moderately positive impact on GDP per capita has been found on iOS smartphone purchases in developing countries. In contrast, the GDP per capita effect on smartphone purchases was significantly positive in developed countries and globally [21]. Given these findings, it is possible that the relationships between GDP per capita and smartphone purchases and choice preferences might be influenced by complex links ignored in previous studies. Some existing research also has focused on the role of Price (PRI), Read-Only Memory (ROM), and Random-Access Memory (RAM) on PUR and SCP [e.g., 19, 20, 25–27]. However, the role of these factors as mediators between GDP per capita and SCP, as well as PUR has not yet been investigated. Moreover, researchers have called for further investigations on PRI, RAM, and ROM across homogenous groups of consumers from various economic settings to further understand SCP and PUR [16]. Accordingly, to fill these gaps and advance research, the present study has considered PRI, RAM, and ROM as potential mediators between GDP per capita (GDP) and SCP and PUR. The current empirical study has been developed based on three theoretical lenses. That is to say, two macroeconomic theories, comprising the Purchasing Power View (PPV) and the Neoclassical Economic Theory (NET), as well as the Behavioral Decision Theory (BDT). The study then tested the research model through SmartPLS-4. The result revealed that the relationships between GDP per capita and SCP and PUR are direct and partially mediated by PRI, ROM, and RAM, which indirectly connect GDP per capita to SCP and PUR.

The structure of the study is as follows. Following the first part, the second part discusses the theoretical context and the development of the hypotheses, and the third part describes the methodology. The fourth part follows with the results and analysis. The discussion, theoretical contributions, limits, and ideas for future research are then presented in the fifth part. The conclusion brings the investigation to a close.

## 2. Theoretical background and hypotheses development

### 2.1. Theoretical background

The use of smartphone devices for daily tasks and their influences on consumers' attitudes and purchase behaviors has been extensively discussed in the literature [1]. However, the number of studies focusing on the purchase of smartphones remains scarce, especially from consumers' economic background perspectives [28]. Researchers have focused more on the use of these devices and their implications on customers' purchasing attitudes and behaviors, ignoring the mechanism and the economic contexts leading to the purchase of these devices [1, 22, 24]. However, one could get a limited understanding of the use of these devices and their implications on customer attitudes without understanding the economic setting that motivates the purchases and choices of these devices [29, 30]. Therefore, understanding the purchase preferences of these devices across different economic contexts constitutes a vital research subject, as it could open up larger fields of study for future research. Similarly, research strongly recommended considering buyer economic settings in smartphone purchase-related studies to better understand buyers' specificities [1], which could assist sellers in creating strategies aligned with customers' preferences [1]. Theoretically, various theories have been utilized within the existing literature to investigate smartphone purchase-related subjects in national frameworks, such as the status quo bias theory, the theory of self-expression and planned behavior, and the cognitive-affective system theory [20]. However, one notes a lack of a theoretical lens explaining, from a macroeconomic perspective, smartphone purchases and choice preferences. Accordingly, we integrated three new theoretical perspectives, i.e., the PWV, NET, and BDT, to examine the relationships between GDP per capita and SCP and PUR.

The purchase preferences of products such as smartphones in an international marketplace, analyzed from the macroeconomic theory point of view, conceptualizes buying as a rational decision-making process based on the macroeconomic environment of the purchaser. In this direction, one can consider that international buyers may select the available options in terms of cost and technical features based on their purchasing power. i.e., the macroeconomic background highlighted by the GDP per capita of such purchasers can be the underpinning of their choice preferences and purchases. Therefore, the macroeconomic setting may be the source of buyers' purchasing power and can influence SCP and PUR to some extent. As, Lu [28] pointed out, buyers always choose smartphone types that match their purchasing power. That purchasing power is measured at each country's level from macroeconomic factors, such as the GDP per capita. From this view, one may assume that the macroeconomic context may influence buyers' SCP and PUR. To support this view, studies show that GDP per capita is an essential indicator for measuring purchasing capacity of a smartphone worldwide [24]. Qi et al. [31] highlighted that countries' macroeconomic indicators such as GDP per capita must be considered when studying CBEC transactions. These macroeconomic indicators give a broad picture of each nation's economic setting to assist comprehend the purchasing power and spending patterns of the people [31]. Therefore, buyers' macroeconomic setting could be considered one of the foundations of their purchase ability and choice preferences.

The macroeconomic perspective is essential because it can explain consumers' purchase ability, the features of their purchases, and their choice preferences since it emphasizes the

buyers' purchasing power. Therefore, we built this study from two macroeconomic theoretical approaches: PPV and NET. According to PPV, the capability to pay determines choice on the market [31]. This view considers consumer choices and purchases as actions arising from purchasing power. However, the consumer's purchasing power derives from his income and the country's wealth, highlighted by factors such as GDP per capita. According to classical economic theory, buying a good depends only on the purchaser's decision [32]. However, this view is debatable in the real world. In this direction, Lu, [28] stated that the purchase also depends on the capability to pay for the desired good, i.e., the buyer's purchasing power. Whatever type of transaction an individual engages in, his purchasing power must be expressed in the form of the ability to buy the desired good [33]. Although purchasing power is a well-known concept in classical economic theory, his conceptualization is somehow overlooked in international purchases [33]. However, it is obvious that individuals or groups differ according to their purchasing power. Moreover, their purchasing power influence their choice preferences [28]. For instance, one knows that a higher purchasing power allows consumers to choose the best qualities [32]. However, this view is overlooked in analyzing economic impacts on transactions because the focus is rather on the willingness to buy and not on the ability to purchase for the desired good [33]. To fill that gap, this study considers purchasing power as one of the underpinnings for SCP and PUR in the CBEC marketplace.

Besides, this study examines SCP and PUR through NET. According to NET, when faced with several product options, consumers always choose rationally to maximize their utility [34]. They always try to take advantage of their money by making the best purchasing decisions [35]. This study considers SCP and purchases as actions that emerge from consumer purchasing power and rationality. Additionally, from the BDT perspective, buyers use context to assess the value of their decision [36]. Mba et al. [37] first pointed out that the context (e.g., the macroeconomic background highlighted in this study) supporting the decision is supposed to influence the decision-making process and the decision-makers judgment. Subsequently, Mba et al. [37] explained that when buyers make a purchase decision, they evaluate the product's relative value based on the context. Moreover, that context significantly influences buyers' judgments during the decision-making process [36]. Fig 1 presents the empirical model.

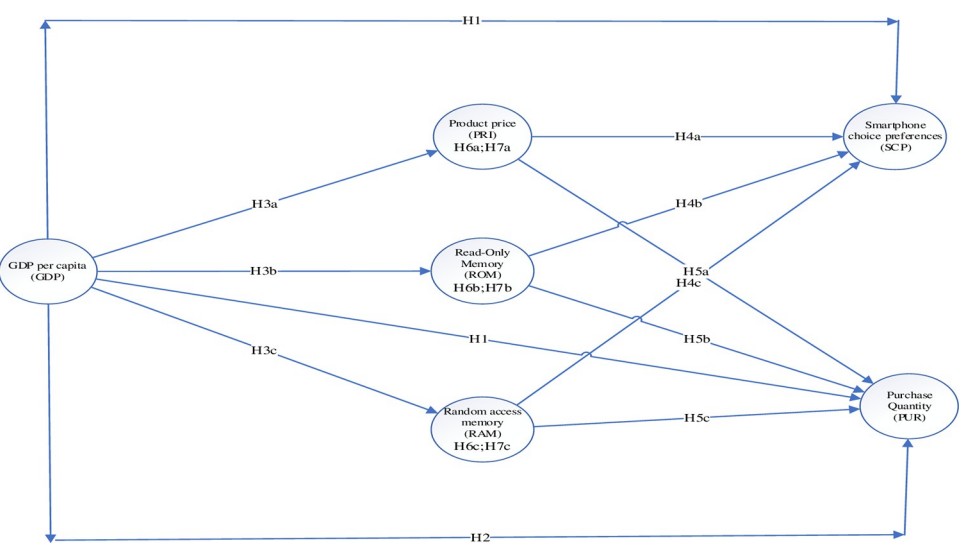

**Fig 1. The empirical model.**

## 2.2. Hypotheses development

### 2.2.1. The effect of GDP per capita on SCP and PUR.

As new digital technologies are gaining momentum, smartphones have become indispensable in all human activities [39]. Today, smartphones are used in exchanges, transport, food orders, communication, transactions, shopping, etc. [39, 40]. In this context, the growing impact of smartphones on human activities is influencing and modifying the practices of economic activity [41].

The economic literature has extensively discussed the concept of the GDP per capita [38]. However, its study in the economic literature is most often focused on economic growth rate forecasting rather than its influence on transactions [16]. In marketing literature, Kröner et al. [39] defined it as the sum of the total gross wealth produced in the economy divided by the number of people in the country at mid-year and able to influence consumer expenses. Since then, the concept of GDP per capita has evolved from economic growth rate forecasting to a strategic factor in making efficient purchase decisions for the citizen in the economy [39]. Therefore, in the marketing literature, the concept of GDP per capita is understood as the sum of the total gross wealth produced in the economy able to impact consumer spending [40]. In the smartphone purchase setting, Jamalova1 and Milán [21] define the concept of GDP per capita as a financial factor enabling consumers to evaluate alternative offers. The concept of GDP per capita thus may influence the choice and expense forms of consumers in the economy. In this current study, SCP refers to smartphone choice preferences according to the available alternative, and PUR refers to the number of smartphones purchased during the transaction by the customer.

In this perspective, research pointed out that a possible relationship between economic growth in a country and smartphone use seems to exist [21]. As, Walton and Nayak [41] highlighted, having a smartphone nowadays may create economic opportunities for users. For that reason, scholars believe that purchases and choices of smartphones could also be influenced by the economic setting of the potential buyer [16, 21]. That is because the type of smartphone a buyer possesses, or purchases can improve that buyer's economic opportunities [41].

Customer preference is the customer's inclination toward a product or service based on tastes and evaluated by its utility. Thus, knowing consumer preference has always been companies' and sellers' goal to meet better their customers' needs [19]. Brands and sellers attempt to gain purchasers' trust by focusing on factors that impact the buyers' decision-making [19]. In doing so, smartphone brands and sellers always wish to be updated about customers' most relevant preferences to meet customer demands and needs [20]. Scholars underpinned that purchase capability in international transactions are based on macroeconomic factors such as GDP per capita [31]. Therefore, they concluded that CBEC activities and purchases could be analyzed using these macroeconomic variables, such as GDP per capita. That is because these variables may enable identifying more active buyers in CBEC [31]. Buyers are generally more attracted to products that match their purchasing power [28]. However, in international transactions, purchasing power is highlighted by macroeconomic indicators such as GDP per capita [42].

Theoretically, various theories have been utilized within the existing literature to investigate subjects related to smartphone choices and purchases in national or local settings [20, 43]. However, concerning the global setting considered in this study, no theoretical lens has been found in the existing literature to explain the link between GDP per capita and SCP as well as PUR. That is due to the limited number of studies on smartphone purchases and related issues [1], especially the lack of studies carried out in a macroeconomic setting [21]. Accordingly, the relationships between GDP per capita and SCP and PUR is explained in this study through the PWV. This theoretical view stipulates that the macroeconomic setting highlighted by the GDP per capita can positively influence customer purchases and preferences. That is so because the

macroeconomic setting is considered one of the foundations of customer purchase ability and choice preferences since it may affect the attitudes and purchase behaviors of the customer. A few studies have studied the link between GDP per capita and smartphone purchases [24]. For instance, [21] highlighted a moderately positive impact of GDP per capita on android smartphone purchases in underdeveloped countries. Meanwhile, the relationship was negative in developed countries and neutral globally [21]. A moderately positive impact on GDP per capita has been highlighted on iOS smartphone purchases in developing countries, while the effect was significantly positive in developed countries and globally [21]. Therefore, contradictory and neutral findings from past research reinforce the need to evaluate in-depth the effect of GDP per capita on smartphone purchases and choice preferences. Hence, this study hypothesizes that:

*H1*: *GDP per capita has a positive effect on SCP*

*H2*: *GDP per capita has a positive effect on PUR*

**2.2.2. The effect of GDP per capita on PRI, ROM, and RAM.** As previously suggested in this research, the contradictory effects of GDP per capita on smartphone purchases and SCP strengthen our misunderstanding of the distinct paths (i.e., direct and indirect channels) that relate GDP per capita to SCP and PUR. Therefore, there is a need to question how GDP per capita affects smartphone purchases and SCP and which factors could act as mediators in that process. Table 1 summarizes the key research that has examined the relationships between macroeconomic context (i.e., GDP per capita, income level, etc.) and smartphone purchases, choices, and diffusions. As shown in Table 1, it is obvious that the GDP capita was not considered enough, but also, the mediators studied (i.e., PRI, ROM, and RAM) in this current study have not been considered yet. Therefore, this study significantly contributes to the literature. It is noteworthy that when the GDP per capita increases or decreases, it can influence citizens' purchasing power for certain goods. Several studies have investigated the impact of GDP per capita on CBEC transactions [31].

He and Wang [44] and Suska [45] highlighted a positive effect of GDP on CBEC transactions. Pan, et al. [46] also suggested a positive effect of GDP per capita on CBEC transactions.

**Table 1. The links between macroeconomic context and smartphone purchases, and choices.**

| Source | Mediators | Macroeconomic variables | Sales | Country | Results |
|---|---|---|---|---|---|
| **Njoh, A. J. (2018)** | None | Human development index (HDI) | Offline sale | 51 African countries | A positive link exists between Human Development Index (HDI) and mobile phone diffusions. |
| **Jamalova and Milán (2019)** | Country Income Level | GDP per capita | Offline sale | Worldwide, developing and developed countries | There is a positive and significant relationship between GDP per capita and smartphone purchases in developed countries. However, the relationship is moderate in developing countries. |
| **Ahmad, et al. (2020)** | None | Smartphone Pricing within country | Offline sale | Pakistan | RAM, and ROM affect positively mobile phone prices. |
| **Jamalova and Constantinovits (2020)** | None | Country Income Level | Offline sale | Worldwide and developing countries | The income level of countries influences smartphone purchases and diffusion. |
| **Odhiambo (2022)** | None | Gini coefficient Index | - | Sub-Saharan African | The Gini coefficient index influences the diffusion of mobile phones in African sub-Saharan countries. |
| **Lechman, E. (2022)** | None | Economic inequalities | - | North Africa and Middle East | The dwindling economic disparities among nations impact the rapid adoption of digital technology like smartphones. |
| **E Lechman, M Popowska (2022)** | None | Economics growth | - | 40 developing countries | Economic growth influences growing ITC product deployment, such as smartphones. |

Price refers to a product's value in terms of the money a customer must spend to own the product [47]. According to Hride et al. [48], the price is one of the product's fundamental values and positively influences customer purchase and decision-making. For that reason, buyers refer to that factor in evaluating alternative offers. Considering price in product evaluation generally contributes to making a good choice [48]. Therefore, consumers tend to be more satisfied when their economic situation matches the product price [26, 49, 50]. Lim et al. [50] pointed out that since consumers may be sensitive to price due to their purchasing capability, focusing on factors that influence price perception is imperative because it allows them to evaluate alternatives and significantly affects customers' purchase and choice preferences.

A smartphone's ROM and RAM are described as quality and specification characteristics that the consumer relies on to assess the merit of the smartphone before making their choice [51]. In this perspective, Bhattacharjee et al. [52] consider ROM and RAM as building blocks of SCP and purchase intention. Pasha and Mu [51] pointed out that Read Only Memory (ROM) and Random Access Memory (RAM) should be considered as a criterion of consumer decision-making. Supporting that view, Bhattacharjee et al. [52] showed that these two features impact SCP and purchase. Therefore, this study considers ROM and RAM features as consumer choice and purchase criteria.

In SCP and PUR, PRI, ROM, and RAM are essential in evaluating alternatives [16]. Those attributes constituent the foundation of SCP and PUR and may be related to buyers' macroeconomic setting [16]. Accordingly, Ahmad et al. [16] hypothesized that countries' macroeconomic backgrounds might influence consumers' choice of smartphone attributes, i.e., PRI, ROM, and RAM. However, they did not have enough data to support their viewpoint. Therefore, they called for more studies on PRI, ROM, and RAM across homogenous groups of consumers from different macroeconomic backgrounds in the international market. Therefore, this study somehow replies to the call of Ahmad et al. [16].

According to the PPV, consumer choices are always related to purchasing power and economic background [53]. In developed countries with higher purchasing power, customers are more willing to purchase products at higher costs than consumers in developing countries because customers from developed countries have a higher purchasing power than those in underdeveloped countries. According to BDT, context is the foundation for assessing the value of a buying decision [54]. That is because the context affects the decision maker's judgment and the decision outcome. Thus, consumers in developed countries with higher purchasing power may be more attracted to buying powerful ROM and RAM at a higher cost (PRI), matching their lifestyle better than consumers in poor countries. Therefore, countries' GDP per capita could affect buyers' choice preferences of PRI, ROM, and RAM. Hence, this study hypothesized the following hypotheses:

*H 3*: *GDP per capita has a positive effect on (a) PRI, (b) ROM, (c) RAM*

**2.2.3. The effect of PRI, ROM, and RAM, on SCP and PUR.** Smartphones' attributes are essential to their success within the marketplace. Ahmad et al. [16] showed that more than 90% of new products are unsuccessful in marketplaces due to attributes that do not match customer preferences and expectations. Some scholars studied the impact of ROM, RAM, and PRI on SCP and PUR [2, 16, 26]. However, the results of those studies were opposed to each other. For instance, with hierarchical regression analysis, Bringula et al. [26] showed a positive influence of PRI on SCP and PUR. However, the effect was insignificant when personal-related variables were incorporated into the analysis. Matthew et al. [3] also showed an insignificant influence of PRI on SCP. However, GSMA [4] highlighted a positive impact of PRI on PUR and SCP. Regarding processing power highlighted by ROM and RAM, Van et al. [5] showed a

positive and significant effect on PUR, while, Agrawal and Gupta [6] discovered that ROM and RAM have an insignificant effect on SCP. However, Paul and Bhukya [7] showed that ROM positively and significantly impacted SCP. Likewise, with the theory of planned behaviour and self-expression, GSMA [4] showed a positive impact of ROM on SCP. Ohme et al. [2] also showed that ROM positively impacts SCP through the cognitive-affective theory.

Although past studies have investigated the impacts of ROM, RAM, and PRI on PUR and SCP, those studies were carried out within national settings through offline marketplaces e.g., [2, 5, 8]. Therefore, those studies' conclusions might not represent a general overview of SCP and PUR globally. However, from the above findings, it is obvious that PRI, ROM, and RAM might influence SCP and PUR. Therefore, we believe that ROM, RAM, and PRI could have significant relationships with SCP and PUR in the CBEC framework.

According to NET, consumers choose rationally to maximize their utilities [9]. They always try to take advantage of their money by making the best purchasing decisions [34]. That theoretical view is well highlighted in the CBEC framework.

Due to some malicious sellers selling fake products, purchasers seem to utilize the price level to verify the product's originality and maximize their utilities. That is because, in online purchases, the quality of products such as smartphones cannot be verified until the purchaser has received them. Therefore, PRI could influence consumer purchase and choice preferences positively. Given the above arguments, the contradictory results of PRI, ROM, and RAM on PUR and SCP in previous studies, and the study's framework, it will be useful to study further PRI, ROM, and RAM on SPC and PUR. Accordingly, the study hypothesized the following hypotheses:

H 4: *(a) PRI, (b) ROM, and (c) RAM have a positive effect on SCP*

H 5: *(a) PRI, (b) ROM, and (c) RAM have a positive effect on PUR*

**2.2.4. Mediators.** The above arguments suggest that smartphone cost and technical features could be valuable criteria for PUR and SPC across different economic settings. We first showed that GDP per capita could positively impact PRI, ROM, and RAM. In a second phase, we highlighted that PRI, ROM, and RAM could positively impact SCP and PUR. Therefore, we believe that PRI, ROM, and RAM have the potential to mediate the relationships between GDP per capita and PUR and SCP. The literature suggests that GDP per capita can have an impact on PRI, ROM and RAM [8]. PRI, ROM, and RAM can, in turn, influence PUR and SCP i.e., [2, 4, 7, 16]. Therefore, the study hypotheses that:

H6: *The relationship between GDP per capita and SCP is positively mediated by (a) PRI, (b) ROM, and (c) RAM.*

H7: *The relationship between GDP per capita and PUR is positively mediated by (a) PRI, (b) ROM, and (c) RAM*

# 3. Methodology

## 3.1. Smartphone market worldwide

According to Kim et al. [20], the smartphone is one of the ITC products experiencing the fastest growth globally. Smartphones are manufactured by about 76 brands worldwide. Out of these 76 brands, China alone accounts about a dozen brands, including Xiaomi, Vivo, Huawei, Oppo, and Coolpad. That is making China one of the most important markets for smartphones. Chinese smartphones entered the international market with a strategy of affordable

prices, which has increased their market shares rapidly across different regions of the world, especially across emerging and developing nations with low-income levels [3]. Therefore, we focused on three Chinese smartphones (i.e., Huawei, Xiaomi, and Oppo) plus Samsung. These smartphones are more affordable for most buyers worldwide, and the most sold out from the CBEC platform we considered in this study.

According to Jamalova and Constantinovits [24], Huawei's market share has been booming for several years. Moreover, his products have reached around 170 countries worldwide [3]. Regarding the Xiaomi brand, it became the leading smartphone brand in the Indian market, with around 28.8% market share [12]. In 2020, the last report of Gartner [55] pointed out that while the global smartphone market has experienced a decline in turnover of about 20%, Xiaomi has been the only brand that experienced growth of 1.4% in 2020. Becoming thus, the third-largest phone brand globally while other significant brands had experienced a decline in their turnover. Samsung is one of the reference brands for smartphones that has been a market leader for several years [1]. Samsung was ranked number 5 among the top global brands in 2020 [1]. However, with the global outbreak, Samsung made only 2% growth [3]. Despite that, Samsung smartphones have received more interest from purchasers worldwide than some global brands such as Apple, especially in Europe [1]. However, the market share of Oppo smartphones has recently decreased due to the COVID-19 pandemic [3].

## 3.2. Data collection and description

The study's sample comprises four smartphone product brands (i.e., Huawei, Xiaomi, Oppo, and Samsung) purchased through the B2C selling mode by international online buyers on a Chinese CBEC platform. These buyers come from 101 countries worldwide.

The data used mainly comes from two different sources. i.e. A private online data store (http://www.100ec.cn/Index/dsb_product.html) and a public source which is the Aliexpress platform. About 1/4 of the data comes from the private online data store, which added 3/4 of the data we downloaded directly from the Aliexpress platform using Octopus Software.

However, it should be noted that the data we got from that private source also came from the Aliexpress platform. The study's data, thus, are made of smartphone purchases from the Aliexpress platform and shipped to consumers outside China. Approximately 19,400 transactions corresponding to 24,043 smartphones purchased were obtained after combining data from the two sources mentioned above. Thus, data collection, cleaning and reorganization were done from June 30, 2022, to July 25, 2022. The whole data can be seen from this link: (https://docs.google.com/spreadsheets/d/1XeqEhDoAgm4r7Mm5yWkF9D31enwGC_4HMtIT4FVYc5Y/edit#gid=1691249773)

AliExpress is one of the leading platforms of B2C sales. As an internationally leading platform for B2C sales, the AliExpress platform sells to about 150 million consumers from 190 countries worldwide with more than 2400 billion visitors every year and over 100 million products [57]. It is made of about 1.1 million e-sellers located in China and outside China (Affiliated stores) [3, 50]. Most of the sellers on the platform are retailers rather than manufacturers, and they obtain products from factories across China for reselling to international buyers [56]. Therefore, this platform constitutes, in somehow, a gateway between smartphones products and global consumers. The data has been grouped in terms of choice of PRI, ROM, RAM, SCP, and PUR, and country. Then, we added the GDP per capita of countries involved in transactions. These GDP per capita are those of 2021 and obtained from the World Bank database [57]. Initially, the dataset was made up of buyers from 109 countries. However, after adding the GDP per capita (GDP), we found the GDP per capita of 8 countries was missing. We then removed transactions of those countries from the dataset and kept transactions from

**Table 2. Variables and summarized statistics.**

| Variables | Mean | STDEV | Min | Max |
|---|---|---|---|---|
| GDP per capita (GDP) | 33464.722 | 25388.307 | 794 | 114705 |
| Product Price (PRI) | 246.392 | 136.732 | 116 | 619 |
| Read-Only Memory (ROM) | 48.889 | 29.468 | 16 | 128 |
| Random-Access Memory (RAM) | 3.643 | 1.209 | 2 | 6 |
| Smartphone Choice Preference (SCP) | - | - | 1 | 17 |
| Purchase quantity (PUR) | 38.163 | 283.596 | 1 | 5772 |
| Brand market share (BMS) | 1.914 | 3.444 | 0 | 34 |
| Brand | SCP | RAM and ROM | SCP | PUR |
| Xiaomi | sku1 | 2GB16GB | 35 | 4391 |
| Samsung | sku2 | 2GB16GB | 20 | 2798 |
| Samsung | sku3 | 2GB32GB | 21 | 1026 |
| Huawei | sku4 | 3GB32GB | 70 | 833 |
| Xiaomi | sku5 | 3GB32GB | 84 | 8360 |
| Samsung | sku6 | 3GB32GB | 90 | 1451 |
| Oppo | sku7 | 3GB32GB | 58 | 1461 |
| Oppo | sku8 | 3GB64GB | 8 | 8 |
| Huawei | sku9 | 4GB32GB | 33 | 405 |
| Huawei | sku10 | 4GB64GB | 21 | 68 |
| Xiaomi | sku11 | 4GB64GB | 40 | 1255 |
| Samsung | sku12 | 4GB64GB | 32 | 151 |
| Oppo | sku13 | 4GB64GB | 16 | 821 |
| Huawei | sku14 | 4GB128GB | 27 | 222 |
| Huawei | sku15 | 6GB64GB | 29 | 144 |
| Xiaomi | sku16 | 6GB64GB | 33 | 548 |
| Oppo | sku17 | 6GB64GB | 13 | 101 |
| Total | - | - | 630 | 24043 |

101 countries. After grouping the 19400 individual choices in terms of countries, 630 transactions were obtained to carry out the study.

Consumers have chosen 17 smartphone categories based on their technical features (RAM and ROM) and prices (PRI). These 17 smartphone types are distributed as follows: 5 for Huawei, 4 for Xiaomi, 4 for Oppo, and 4 for Samsung. Smartphone Choice Preference (SCP) and Smartphone purchase quantity (PUR) are the dependent variables, while GDP per capita represents the independent variable. PRI, RAM, and ROM were mediators between GDP per capita and the two dependent variables (i.e., SCP and PUR). The dataset also contains the BMS (Brand Market Share) within each country involved in the purchases. However, we did not incorporate that variable into the model. These market shares are those of 2021 and were obtained from the Statcounter Globalstats database [58]. Table 2 presents variables, definitions and trends.

## 4. Results and data analysis

### 4.1. Discriminant validity

To test the study's hypotheses, we applied the technique of SEM through SmartPLS-4, as it is convenient for analyzing consumers' choice preferences [59]. We started our analysis by assessing the discriminant validity between constructs (latent variable), as each construct in our inner model comprises one single attribute. Therefore, there is no need to evaluate the

**Table 3. Discriminant validity using the criterion by Fornell & Larcker HTMT method.**

|  | GDP | PRI | PUR | RAM | ROM | SCP |
|---|---|---|---|---|---|---|
| GDP | 1 | 0.305 | 0.097 | 0.35 | 0.311 | 0.384 |
| PRI | 0.305 | 1 | 0.054 | 0.85 | 0.743 | 0.559 |
| PUR | -0.097 | -0.054 | 1 | 0.086 | 0.076 | 0.088 |
| RAM | 0.35 | 0.85 | -0.086 | 1 | 0.698 | 0.798 |
| ROM | 0.311 | 0.743 | -0.076 | 0.698 | 1 | 0.586 |
| SCP | 0.384 | 0.559 | -0.088 | 0.798 | 0.586 | 1 |

Note: Values in Italic and the Diagonal represent the square roots of the AVE (average variance extracted). Below the diagonal elements are the values of Fornell-Licker Criterion and Above the diagonal are the Heterotrait- Monotrait (HTMT) Ration values.

constructs' convergent validity and reliability. The discriminant validity of constructs was assessed by comparing correlations among variables through the Fornell-Larcker Criterion and heterotrait–monotrait ratio of correlations [59]. All of the results obtained from heterotrait–monotrait ratio of correlations were less than the threshold of 0.85. The values of Fornell-Larcker Criterion showing that the Average Variance-Extracted's (AVE) square roots equal to 1 are higher than all the correlation values between the constructs. Both tests are presented in Table 3. Additionally, the model's goodness and predictive capability are evaluated based on $R^2$ and $Q^2$ values [60]. The values of $R^2$ and $Q^2$ are over 0.1., and 0, respectively, showing that the model has predictive relevance. Both tests are presented in Table 4.

## 4.2. The link between GDP per capita and PUR and SCP

The hypotheses H1 and H2 were ascertained if GDP per capita positively and directly impacts SCP and PUR, respectively. Results revealed that GDP per capita impact positively on SCP ($\beta$ = 0.108, t = 4.635, p = 0.000), and had a negative effect on PUR ($\beta$ = -0.073, t = 3.76, p = 0.000). Hence, H1 is supported but H2 failed to be supported. Next, our results showed a significantly

**Table 4. Structural model results.**

| Hypothesis | GDP -> SCP and PUR | β | STDEV | T | P | 2.50% | 97.50% |
|---|---|---|---|---|---|---|---|
| H1 | GDP -> SCP | 0.108 | 0.023 | 4.635 | 0 | 0.064 | 0.154 |
| H2 | GDP -> PUR | -0.073 | 0.019 | 3.76 | 0 | -0.103 | -0.049 |
| H3a | GDP -> PRI | 0.305 | 0.036 | 8.576 | 0 | 0.234 | 0.373 |
| H3b | GDP -> ROM | 0.311 | 0.035 | 8.792 | 0 | 0.243 | 0.382 |
| H3c | GDP -> RAM | 0.35 | 0.033 | 10.55 | 0 | 0.286 | 0.414 |
| H4a | PRI -> SCP | -0.541 | 0.078 | 6.889 | 0 | -0.709 | -0.401 |
| H4b | ROM -> SCP | 0.201 | 0.049 | 4.109 | 0 | 0.089 | 0.285 |
| H4c | RAM -> SCP | 1.079 | 0.061 | 17.764 | 0 | 0.97 | 1.206 |
| H5a | PRI -> PUR | 0.1 | 0.046 | 2.167 | 0.03 | 0.027 | 0.194 |
| H5b | ROM -> PUR | -0.049 | 0.02 | 2.416 | 0.016 | -0.087 | 0 |
| H5c | RAM -> PUR | -0.111 | 0.053 | 2.076 | 0.038 | -0.223 | -0.045 |
|  | The model's Goodness | R2 | $Q^2$ |  |  |  |  |
|  | SCP | 0.718 | 0.702 |  |  |  |  |
|  | PUR | 0.115 | 0.015 |  |  |  |  |
|  | PRI | 0.193 | 0.089 |  |  |  |  |
|  | ROM | 0.197 | 0.093 |  |  |  |  |
|  | RAM | 0.123 | 0.117 |  |  |  |  |

positive impact of GDP per capita, respectively, on the choice of PRI ($\beta$ = 0.305, t = 8.576, p = 0.000), ROM ($\beta$ = 0.311, t = 8.792, p = 0.000), and RAM ($\beta$ = 0.35, t = 10.55, p = 0.000), supporting thus, H3a, H3b, and H3c. The study also revealed that PRI affects significantly and negatively SCP ($\beta$ = -0.541, t = 6.889, p = 0.000). However, ROM ($\beta$ = 0.201, t = 4.109, p = 0.000), and RAM ($\beta$ = 1.079, t = 17.764, p = 0.000) positively and significantly impact SCP. Hence, H4b and H4c are supported, but H4a failed to be supported. Additionally, the study showed that PRI significantly and positively impacts PUR ($\beta$ = 0.1, t = 2.167, p = 0.030), supporting H5a. However, it showed that ROM ($\beta$ = -0.049, t = 2.416, p = 0.016), and RAM ($\beta$ = -0.111, t = 2.076, p = 0.038), negatively impact PUR. Hence, H5b and H5c failed to be supported. Tables 4 and 5 present the summarized results of the hypotheses testing. To be confident about the model's strength, 5,000 resamples have generated 95% confidence intervals, presented in Table 4, showing a significant link. The results of all assessments are shown in Tables 4–6.

### 4.3. Mediators' assessment result

Mediation testing was carried out to evaluate PRI, RAM, and ROM's mediating roles between GDP per capita and SCP and PUR. First of all, the analysis results in Table 4 revealed a significantly negative and partial mediating role of PRI ($\beta$ = -0.165, t = 5.181, p = 0.000) between GDP per capita and SCP. However, it has been revealed that ROM ($\beta$ = 0.063, t = 4.059, p = 0.000), and RAM ($\beta$ = 0.378, t = 9.104, p = 0.000) have a positive and partial mediating influence between GDP per capita and SCP; thus, H6b and H6c are supported but H6a failed to be supported. Next, the study showed that PRI ($\beta$ = 0.03, t = 1.994, p = 0.046) has a significantly positive and partial mediating role between GDP per capita and PUR. However, the analysis revealed that ROM ($\beta$ = -0.015, t = 2.361, p = 0.018), and RAM ($\beta$ = -0.039, t = 1.998, p = 0.046) have significantly negative and partial mediating roles between GDP per capita and PUR. Hence, H4b and H4c are supported, but H7a is not.

## 5. Discussion

Our results revealed that the direct impact of GDP on PUR was significantly negative. That result is somehow opposed to the findings of Jamalova and Milán [21] and Jamalova and

**Table 5. Mediation analysis.**

| Total Effect of GDP -> SCP and GDP -> PUR | | | | |
|---|---|---|---|---|
| | β | T | P | |
| GDP -> SCP | 0.384 | 11.956 | 0.000 | |
| GDP -> PUR | -0.097 | 3.63 | 00.000 | |
| Direct Effect of GDP -> SCP and GDP -> PUR | | | | |
| β | T | P | | |
| 0.108 | 4.635 | 0.000 | | |
| -0.073 | 3.76 | 0.000 | | |
| Indirect Effect of GDP -> SCP and GDP -> PUR | | | | |
| | | β | T | P |
| H6a | GDP -> PRI -> SCP | -0.165 | 5.181 | 0.000 |
| H6b | GDP -> ROM -> SCP | 0.063 | 4.059 | 0.000 |
| H6c | GDP -> RAM -> SCP | 0.378 | 9.104 | 0.000 |
| H7a | GDP -> PRI -> PUR | 0.030 | 1.994 | 0.046 |
| H7b | GDP -> ROM -> PUR | -0.015 | 2.361 | 0.018 |
| H7c | GDP -> RAM -> PUR | -0.039 | 1.998 | 0.046 |

**Table 6. Relationships and hypotheses.**

| Relationships | Hypotheses | Results |
| --- | --- | --- |
| GDP -> SCP | H1 | supported |
| GDP -> PUR | H2 | Not supported |
| GDP-> PRI, ROM, and RAM | | |
| GDP -> PRI | H3a | Supported |
| GDP -> ROM | H3b | Supported |
| GDP -> RAM | H3c | Supported |
| PRI, ROM, and RAM -> SCP | | |
| PRI -> SCP | H4a | Not supported |
| ROM -> SCP | H4b | Supported |
| RAM -> SCP | H4c | Supported |
| PRI, ROM, and RAM -> PUR | | |
| PRI -> PUR | H5a | Supported |
| ROM -> PUR | H5b | Not supported |
| RAM -> PUR | H5c | Not supported |
| GDP-> MEDIATORS | | |
| GDP -> PRI -> SCP | H6a | Not supported |
| GDP -> ROM -> SCP | H6b | Supported |
| GDP -> RAM -> SCP | H6c | Supported |
| GDP -> PRI -> PUR | H7a | Supported |
| GDP -> ROM -> PUR | H7b | Not supported |
| GDP -> RAM -> PUR | H7c | Not supported |

Constantinovits [24]. Jamalova and Milán [21] found that the impact of GDP per capita on android purchases was insignificant globally, while the effect was negative in developed countries and positive in underdeveloped countries. Jamalova and Constantinovits [24] also found a positive impact of the economic level on PUR globally. However, depending on economic contexts, the significance of that effect was diverse. In developing countries, that impact was less significant compared to other economic contexts [24].

Regarding PRI, RAM, and ROM choice, our results revealed a positive impact of GDP per capita on PRI, ROM, and RAM significantly. These outcomes might suggest that in purchases and SCP, consumers are looking for features that match their macroeconomic or economic setting [16]. This observation aligns with the conjecture of Ahmad et al. [16]. Ahmad et al. [16] pointed out that purchasers prefer buying smartphones with RAM and ROM matching their economic settings. The study also revealed that PRI negatively impacted SCP. That outcome is contrary to the results of Wong [27] and Kim et al. [61]. This result is understandable because most purchased smartphones had RAM of 3GB and a ROM of 32GB. That type of smartphone had relatively low prices compared to other smartphones, such as RAM = 4GB ROM = 64GB. The study also showed that PRI positively and significantly affects PUR. Bringula et al. [26] also reported similar results showing that PRI positively and significantly impacts purchases when personal-related variables are not considered. Pinto et al. [19] supported that observation by pointing out that most purchasers buy smartphones based on price alone. The negative results of PRI on SCP and that of PRI on PUR (positive) could be explained from two perspectives: The Smartphone Choice Preference (SCP) and Smartphone Purchase Quantity (PUR) perspectives. That is to say, when purchasers have to choose among several alternatives (SCP) for personal use, they seem to choose cheaper products. However, when purchasing several products at once (Purchasing in Quantity), buyers choose the expensive ones. That is

understandable because, in CBEC, purchasers buying in quantities are usually resellers. They care about the product's quality. Therefore, they may utilize price features to verify product quality due to malicious e-sellers selling fake products instead of the original ones. Hence, buyers may assume the products are not original when the price is too low. RAM and ROM play essential roles in PUR and SCP as they are crucial in data storing and the running of smartphone applications Ahmed et al. [16]. Our results showed that ROM and RAM positively and significantly impact SCP. These results support previous studies showing ROM and RAM's importance in SCP [19, 20, 61]. The study has shown that ROM and RAM significantly negatively impact PUR. That result contradicts the finding of Liu and Liang [18]. They showed insignificant relationships between RAM, ROM, and PUR. RAM and ROM's contradictory impact on PUR and SCP could be understood because most consumers' purchases and choices were focused on RAM = 3GB and ROM = 32GB, which have low RAM and ROM capability compared to other types, such as RAM = 4GB and ROM = 64GB. It can be concluded that buyers prefer higher RAM and ROM when choosing a smartphone for personal use. However, when buyers want to purchase more products at once, perhaps for reselling purposes, they prefer to purchase smartphones with average RAM (3GB) and ROM (32GB) capability.

The study revealed a positive and partial mediating role of ROM and RAM between GDP per capita and SCP. However, the study highlighted a negative and partial mediating role of ROM and RAM between GDP per capita and PUR. That is to say, the higher the GDP is, the more buyers choose smartphones with high ROM and RAM levels (smartphones with high treatment capability). However, the higher the GDP per capita is, the lower the number of smartphone purchases from that country. According to these results, it can be concluded that CBEC buyers prefer to choose the ROM and RAM of smartphones in line with their macroeconomic background when it comes to choices (SCP) for personal use. However, as already stated, the economic background is not considered when purchasing a higher number of smartphones at once. That can be explained by the fact that the smartphone purchases in quantity within the CBEC context come more from less developed economic countries devoid of smartphone manufacturing industries. The buyers who purchase higher quantities at once on CBEC platforms are resellers that buy from the platforms and resell them on the local markets. That could explain why GDP per capita, PRI, ROM, and RAM do not seem to have the same impact on SCP and PUR.

## 5.1. Theoretical contributions and practical implications for e-sellers, brand managers, and consumers

In terms of the study's theoretical contributions, this study reinforces our understanding of how smartphone technical characteristics (ROM and RAM) and pricing (PRI) influence choice preferences and purchase quantities in the smartphone selling industry. Therefore, the primary theoretical contribution of this study is a better understanding of the mechanism of the influence of GDP per capita on SCP and PUR via PRI, ROM, and RAM, which revealed that the linkages between GDP per capita, SCP, and PUR are not only direct but also partially mediated by other factors that indirectly link GDP per capita, SCP, and PUR. Therefore, this study's findings supplement and improve the small number of existing studies that empirically examined the influence of GDP per capita on SCP and PUR in the smartphone selling industry. The present article's outcome suggests that buyers' choice and purchase of smartphones are not homogeneous and depend on economic contexts. That being so, it seems more appropriate for future studies to move beyond the national contexts to offer conceptual and empirical models in the smartphone selling industry that are globally oriented.

The study provides a robust theoretical contribution to the literature because there is limited research on the relationships between GDP per capita and SCP and PUR through PRI, ROM, and RAM as mediators. Almost no studies have focused on PRI, ROM, and RAM as mediators in analyzing the impact of GDP per capita on SCP and PUR. In doing so, this study responds to the calls of two groups of researchers: First, the study replies to the call of Paul and Bhukya [7], who called for more research on smartphone purchase preferences [1] in order to better understand and address the needs of consumers [7]. Second, the study also replies to the call of Ahmad et al. [16], who called for more studies on PRI, RAM, and ROM across homogenous groups of consumers across various economic backgrounds in the international market to understand further the impact of PRI, RAM, and ROM on SCP and purchases. Moreover, using PWV, NET, and BDT as a theoretical lens, the study expanded the theoretical backgrounds utilized to study smartphone purchase and choice preferences in previous studies.

Given buyers' different economic backgrounds, it is difficult for e-sellers of CBEC marketplaces to predict buyers' potential choice preferences and purchases for a given smartphone. In such a context, this current study can assist practitioners in understanding how GDP per capita or macroeconomic indicators can drive purchases and SCP and provide valuable information for the smartphone-selling industry. Smartphone brands and e-sellers must adapt their marketing strategies to the consumer macroeconomic context to get a place on the rostrum of consumer preferences [1].

Additionally, in considering the findings of this study, smartphone companies could better understand the importance of designing and proposing different technical features of smartphones in line with the macroeconomic settings of buyers to generate more purchases and profits [28]. That could also have a significant implication for recommendation systems studies in the case of international online sales. In international online shopping, recommendation systems usually recommend products to possible buyers without considering the consumers' macroeconomic context. Therefore, this study contributes to an unsolved theoretical and management issue. That is to say, how to map out SCP and PUR through macroeconomic settings to assist e-sellers and smartphone companies' marketing activities in anticipating SCP and PUR. Past research's inability to provide a theoretical and empirical framework to examine linkages between macroeconomic contexts and SCP and PUR may be part of the reason for this unsolved theoretical and managerial challenge. Therefore, the result of this study could assist in filling that gap and add another dimension to the factors utilized in the recommendation systems, thus strengthening CBEC recommendation system studies. Improvement of recommendation systems studies will facilitate the buying and decision-making process by helping customers make the right choices in line with their economic context.

Furthermore, the current study's findings can be utilized for smartphone marketing campaigns directed toward international buyers. Other advantages could be the ease of managing transactions and purchases by e-sellers. For instance, e-sellers might adapt their selling strategy to consumers' choice preferences and purchase behaviors based on discoveries made in this study by developing a dual selling strategy. The first may be based on consumer choice preferences, while the second may be based on consumers' purchasing quantities. So, when a buyer shows the desire to buy a single smartphone, the seller should offer the buyer a smartphone according to his macroeconomic background. However, when a buyer shows the desire to purchase in quantities, the e-seller should use the second strategy. That strategy is to offer possible smartphone buyers with average RAM and ROM capabilities while carefully managing the price. Because this type of buyer is likely to be a reseller rather than an end-user. This study provides practitioners, e-sellers, and smartphone brands with an opportunity to update their marketing strategies, smartphone costs, and technical features to buyers' macroeconomic settings.

### 5.2. Limitations and future research directions

The current study is somewhat limited. This current research focused only on one macroeconomic factor, two smartphone technical features (RAM and ROM), and PRI. Since consumers worldwide vary significantly in cultural features, psychological characteristics, and social factors, other macroeconomic variables and smartphone technical features may be integrated to further study SCP and PUR. Second, the research has tested hypotheses using secondary data from a single CBEC platform. Future studies may focus on cross-section, primary, or longitudinal data that could assist in assessing the changes of SCP and PUR through different macroeconomic backgrounds. Alternatively, researchers may consider downloading data from various CBEC platforms to verify our findings. Third, future research might look at the mediating role of other factors, such as the influence of shipping prices on the linkage between GDP per capita and smartphone purchases.

Although SCP and PUR were affected by PRI, ROM, and RAM, they were found to be negative mediators in some conditions. That is to say, PRI was found to be a negative mediator between GDP per capita and SCP, while ROM and RAM were found to be negative mediators between GDP and PUR in this current empirical model. Therefore, other future studies could integrate them as potential mediators and keep examining their role in similar empirical or new conceptual models. Then, make conclusions about the role of those variables in the relationship between GDP per capita (or other macroeconomic variables), smartphone buying behavior, and choice preferences. Fourth, no moderating factors were examined in the current study. However, the buying and choice of smartphones in the CBEC context is a decision that might be associated with other factors, such as product brands' availability in the local market of each country. If the product can be found easily in the local markets, consumers will less frequently buy it on CBEC marketplaces. Future studies may thus look at the moderating impact of local market product availability.

## 6. Conclusion

The first finding in this paper is that the relationships between GDP per capita and SCP and PUR are direct and partially mediated by PRI, ROM, and RAM that connect GDP per capita to SCP and PUR indirectly. That is to say, positive and partial mediating roles of ROM and RAM have been found between GDP per capita and SCP, while positive and partial mediating roles of PRI are also found between GDP per capita and PUR. In doing so, the study substantiates the view of existing research that has found a significant impact of GDP per capita on purchases made through CBEC settings. Additionally, the results revealed that although GDP per capita affected SCP and PUR, the direction of the impact was negative on PUR and positive on SCP (Table 5). Moreover, PRI was found to be a negative mediator between GDP and SCP, while ROM and RAM were negative mediators between GDP and PUR in our conceptual model (Table 5).

## Supporting information

**S1 File.**
(RAR)

## Acknowledgments

The authors sincerely thank the editors, reviewers, as well as all of the anonymous for the valuable help and contributions.

## Author Contributions

**Data curation:** Karamoko N'da.

**Formal analysis:** Karamoko N'da.

**Funding acquisition:** Jiaoju Ge.

**Methodology:** Karamoko N'da.

**Resources:** Jiaoju Ge.

**Supervision:** Jiaoju Ge, Steven Ji-Fan Ren, Jia Wang.

**Validation:** Jiaoju Ge, Steven Ji-Fan Ren.

**Writing – original draft:** Karamoko N'da.

**Writing – review & editing:** Jiaoju Ge, Steven Ji-Fan Ren.

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
