## [Decision Letter · Decision Letter 0]

10 Jan 2023

PONE-D-22-29643What Matters for International consumers’ Choice preferences for Smartphones: Evidence from a Cross-border Ecommerce PlatformPLOS ONE

Dear Dr. Karamoko N'da,

Thank you for submitting your manuscript to PLOS ONE. After careful consideration, we feel that it has merit but does not fully meet PLOS ONE’s publication criteria as it currently stands. Therefore, we invite you to submit a revised version of the manuscript that addresses the points raised during the review process.

We look forward to receiving your revised manuscript.

Kind regards,

Vincenzo Basile, PhD

Academic Editor

PLOS ONE

Journal Requirements:

“No The funders had no role in study design, data collection and analysis, decision to publish, or preparation of the manuscript”

Reviewers' comments:

Reviewer's Responses to Questions

**Comments to the Author**

1. Is the manuscript technically sound, and do the data support the conclusions?

Reviewer #1: Yes

Reviewer #2: Yes

2. Has the statistical analysis been performed appropriately and rigorously? 

Reviewer #1: I Don't Know

Reviewer #2: Yes

3. Have the authors made all data underlying the findings in their manuscript fully available?

Reviewer #1: Yes

Reviewer #2: No

4. Is the manuscript presented in an intelligible fashion and written in standard English?

Reviewer #1: Yes

Reviewer #2: Yes

5. Review Comments to the Author

Reviewer #1: The authors have done a lot of work, examining the consumers' inclinations to buy smartphones in a multi-threaded manner.

The article is well written with the correct structure of this type of work.

Nevertheless, sometimes the excess of content is overwhelming, often some content is duplicated (especially in the Introduction and Theoretical background and hypotheses development sections), which limits the clarity of the presented arguments.

I also believe that authors should write certain sentences differently, e.g.:

"As [28] pointed out (...)" - I believe that the surname or surnames of the authors of the cited publication should be quoted

"[31] highlighted that (...)" - a specific surname (names) should also be indicated here, e.g. "Qi and others [31] highlighted that (...)".

The text should be verified in this regard.

Reviewer #2: The article presents the results of an interdisciplinary attempt to combine strictly economic areas, i.e. a macroeconomic perspective, including GDP per capita within approaches of PPV and NET, with the purchase-related areas within the status quo bias theory, the theory of self-expression and planned behavior, and the cognitive-affective system theory.

The macroeconomic perspective is one of the foundations of customers' purchasing abilities and choice preferences, because it can influence customer attitudes and behavior. Thus, the positive impact of PKP per capita on smartphone choice preferences and purchase quantities is obvious, widely described in the literature, so I would indicate the adopted hypotheses as unnecessary. On the other hand, the hypotheses related to price, ROM and RAM are actually an added value to the current literature in this area.

Due to the fact that two areas of science governed by their own laws are being combined at the junction, in order to avoid controversies and even more - mistakes, it seems necessary:

- indication of the definitions of all the main constructs adopted in the research; despite the fact that the concepts are not advanced, their juxtaposition in interdisciplinary terms may raise doubts,

- indication of the scope of the concept of consumers purchase behaviors, which is interpreted differently in the literature; among the dominant theoretical trends, depending on the scope adopted in the research procedure adopted, the concept may have a different reference to the results,

- the source of access to the data used in the research, i.e. the chosen Cross-Border E-Commerce platform, was indicated too enigmatically; please specify what data and how they were made available, which are publicly available, how they are collected, since according to the authors they are a reliable source.

6. PLOS authors have the option to publish the peer review history of their article (what does this mean?). If published, this will include your full peer review and any attached files.

Reviewer #1: No

Reviewer #2: No

---

## [Author Response · Author response to Decision Letter 0]

15 Apr 2023

Reviewer #1: The authors have done a lot of work, examining the consumers' inclinations to buy smartphones in a multi-threaded manner. The article is well written with the correct structure of this type of work. Nevertheless, sometimes the excess of content is overwhelming, often some content is duplicated (especially in the Introduction and Theoretical background and hypotheses development sections), which limits the clarity of the presented arguments. 

We sincerely appreciate your positive comments on the correct structure of the work. Regarding duplicated contents, we have tried our best to check and eliminate the duplicated contents in the Introduction and Theoretical background and hypotheses development sections, as you have suggested.

Reviewer #1: I also believe that authors should write certain sentences differently, e.g.: "As [28] pointed out (...)" - I believe that the surname or surnames of the authors of the cited publication should be quoted "[31] highlighted that (...)" - a specific surname (names) should also be indicated here, e.g. "Qi and others [31] highlighted that (...)". The text should be verified in this regard.

Thanks for your constructive comments and suggestions. We corrected the citations as you suggested by indicating the surname (names).

Reviewer #2: The article presents the results of an interdisciplinary attempt to combine strictly economic areas, i.e. a macroeconomic perspective, including GDP per capita within approaches of PPV and NET, with the purchase-related areas within the status quo bias theory, the theory of self-expression and planned behavior, and the cognitive-affective system theory. The macroeconomic perspective is one of the foundations of customers' purchasing abilities and choice preferences, because it can influence customer attitudes and behavior. Thus, the positive impact of GDP per capita on smartphone choice preferences and purchase quantities is obvious, widely described in the literature, so I would indicate the adopted hypotheses as unnecessary. On the other hand, the hypotheses related to price, ROM and RAM are actually an added value to the current literature in this area. 

Thanks for your constructive and positive comments. Although the positive impact of GDP per capita on smartphone choice preferences and purchase quantities is obvious and widely described in the literature, we believe that it was necessary to highlight before adding other variables given the dynamic economic changes and consumer tastes worldwide. We sincerely thank you for this positive comment on the manuscript.

Reviewer #2: Due to the fact that two areas of science governed by their own laws are being combined at the junction, in order to avoid controversies and even more - mistakes, it seems necessary:

- indication of the definitions of all the main constructs adopted in the research; despite the fact that the concepts are not advanced, their juxtaposition in interdisciplinary terms may raise doubts. - indication of the scope of the concept of consumers purchase behaviors, which is interpreted differently in the literature; among the dominant theoretical trends, depending on the scope adopted in the research procedure adopted, the concept may have a different reference to the results. 

Thanks for the kind suggestion. We defined all the main constructs adopted in the research by taking into account the scope of the concepts in the revised manuscript, as you suggested, as follows: 

The economic literature has extensively discussed the concept of the GDP per capita [38]. However, its study in the economic literature is most often focused on economic growth rate forecasting rather than its influence on transactions [16]. In marketing literature, Kröner. et al. [39] defined it as the sum of the total gross wealth produced in the economy divided by the number of people in the country at mid-year and able to influence consumer expenses. Since then, the concept of GDP per capita has evolved from economic growth rate forecasting to a strategic factor in making efficient purchase decisions for the citizen in the economy [39]. Therefore, in the marketing literature, the concept of GDP per capita is understood as the sum of the total gross wealth produced in the economy able to impact consumer spending [40]. In the smartphone purchase setting, Jamalova1 and Milán [21] define the concept of GDP per capita as a financial factor enabling consumers to evaluate alternative offers. The concept of GDP per capita thus may influence the choice and expense forms of consumers in the economy. In this current study, SCP refers to smartphone choice preferences according to the available alternative, and PUR refers to the number of smartphones purchased during the transaction by the customer.

Price refers to a product's value in terms of the money a customer must spend to own the product [47]. According to Hride et al. [48], the price is one of the product's fundamental values and positively influences customer purchase and decision-making. For that reason, buyers refer to that factor in evaluating alternative offers. Considering price in product evaluation generally contributes to making a good choice [48]. Therefore, consumers tend to be more satisfied when their economic situation matches the product price [49-50-51]. Lim et al. [51] pointed out that since consumers may be sensitive to price due to their purchasing capability, focusing on factors that influence price perception is imperative because it allows them to evaluate alternatives and significantly affects customers' purchase and choice preferences.

A smartphone's ROM and RAM are described as quality and specification characteristics that the consumer relies on to assess the merit of the smartphone before making their choice [52]. In this perspective, Bhattacharjee et al. [53] consider ROM and RAM as building blocks of SCP and purchase intention. Pasha and Mu [52] pointed out that Read Only Memory (ROM) and Random Access Memory (RAM) should be considered as a criterion of consumer decision-making. Supporting that view, Bhattacharjee et al.[53] showed that these two features impact SCP and purchase. Therefore, this study considers ROM and RAM features as consumer choice and purchase criteria.

Reviewer #2: - the source of access to the data used in the research, i.e. the chosen Cross-Border E-Commerce platform, was indicated too enigmatically; please specify what data and how they were made available, which are publicly available, how they are collected, since according to the authors they are a reliable source.

Thanks for your helpful comments in revising the manuscript. For convenience, we tried our best to clarify the data source. We answered your questions about how they were made available, which part of the data is publicly available, and how the data was collected. We have explained it as follows:

The study's sample comprises four smartphone product brands (i.e., Huawei, Xiaomi, Oppo, and Samsung) purchased through the B2C selling mode by international online buyers on a Chinese CBEC platform. These buyers come from 101 countries worldwide. 

The data used mainly comes from two different sources. i.e. A private online data store (http://www.100ec.cn/Index/dsb_product.html) and a public source which is the Aliexpress platform. About 1/4 of the data comes from the private online data store, which added 3/4 of the data we downloaded directly from the Aliexpress platform using Octopus Software.

However, it should be noted that the data we got from that private source also came from the Aliexpress platform. The study's data, thus, are made of smartphone purchases from the Aliexpress platform and shipped to consumers outside China. Approximately 19,400 transactions corresponding to 24,043 smartphones purchased were obtained after combining data from the two sources mentioned above. Thus, data collection, cleaning and reorganization were done from June 30, 2022, to July 25, 2022. The whole data can be seen from this link: (https://docs.google.com/spreadsheets/d/1XeqEhDoAgm4r7Mm5yWkF9D31enwGC_4HMtIT4FVYc5Y/edit#gid=1691249773) 

AliExpress is one of the leading platforms for B2C sales. As an internationally leading platform for B2C sales, the AliExpress platform sells to about 150 million consumers from 190 countries worldwide, with more than 2400 billion visitors every year and over 100 million products [58]……

---

## [Editor Report · Decision Letter 1]

18 Apr 2023

PONE-D-22-29643R1What Matters for International consumers’ Choice preferences for Smartphones: Evidence from a Cross-border Ecommerce PlatformPLOS ONE

Dear Dr. Karamoko N'da,

Thank you for submitting your manuscript to PLOS ONE. After careful consideration, we feel that it has merit but does not fully meet PLOS ONE’s publication criteria as it currently stands. Therefore, we invite you to submit a revised version of the manuscript that addresses the points raised during the review process.

ACADEMIC EDITOR: Please provide a final paper with all revisions made and I recommend an additional check on plagiarism and/or compliance with the Journal's guidelines.

We look forward to receiving your revised manuscript.

Kind regards,

Vincenzo Basile, PhD

Academic Editor

PLOS ONE
---

## [Author Response · Author response to Decision Letter 1]

20 Apr 2023

Reviewer #1: The authors have done a lot of work, examining the consumers' inclinations to buy smartphones in a multi-threaded manner. The article is well written with the correct structure of this type of work. Nevertheless, sometimes the excess of content is overwhelming, often some content is duplicated (especially in the Introduction and Theoretical background and hypotheses development sections), which limits the clarity of the presented arguments. 

We sincerely appreciate your positive comments on the correct structure of the work. Regarding duplicated contents, we have tried our best to check and eliminate the duplicated contents in the Introduction and Theoretical background and hypotheses development sections, as you have suggested.

Reviewer #1: I also believe that authors should write certain sentences differently, e.g.: "As [28] pointed out (...)" - I believe that the surname or surnames of the authors of the cited publication should be quoted "[31] highlighted that (...)" - a specific surname (names) should also be indicated here, e.g. "Qi and others [31] highlighted that (...)". The text should be verified in this regard.

Thanks for your constructive comments and suggestions. We corrected the citations as you suggested by indicating the surname (names).

Reviewer #2: The article presents the results of an interdisciplinary attempt to combine strictly economic areas, i.e. a macroeconomic perspective, including GDP per capita within approaches of PPV and NET, with the purchase-related areas within the status quo bias theory, the theory of self-expression and planned behavior, and the cognitive-affective system theory. The macroeconomic perspective is one of the foundations of customers' purchasing abilities and choice preferences, because it can influence customer attitudes and behavior. Thus, the positive impact of GDP per capita on smartphone choice preferences and purchase quantities is obvious, widely described in the literature, so I would indicate the adopted hypotheses as unnecessary. On the other hand, the hypotheses related to price, ROM and RAM are actually an added value to the current literature in this area. 

Thanks for your constructive and positive comments. Although the positive impact of GDP per capita on smartphone choice preferences and purchase quantities is obvious and widely described in the literature, we believe that it was necessary to highlight before adding other variables given the dynamic economic changes and consumer tastes worldwide. We sincerely thank you for this positive comment on the manuscript.

Reviewer #2: Due to the fact that two areas of science governed by their own laws are being combined at the junction, in order to avoid controversies and even more - mistakes, it seems necessary:

- indication of the definitions of all the main constructs adopted in the research; despite the fact that the concepts are not advanced, their juxtaposition in interdisciplinary terms may raise doubts. - indication of the scope of the concept of consumers purchase behaviors, which is interpreted differently in the literature; among the dominant theoretical trends, depending on the scope adopted in the research procedure adopted, the concept may have a different reference to the results. 

Thanks for the kind suggestion. We defined all the main constructs adopted in the research by taking into account the scope of the concepts in the revised manuscript, as you suggested, as follows: 

The economic literature has extensively discussed the concept of the GDP per capita [38]. However, its study in the economic literature is most often focused on economic growth rate forecasting rather than its influence on transactions [16]. In marketing literature, Kröner. et al. [39] defined it as the sum of the total gross wealth produced in the economy divided by the number of people in the country at mid-year and able to influence consumer expenses. Since then, the concept of GDP per capita has evolved from economic growth rate forecasting to a strategic factor in making efficient purchase decisions for the citizen in the economy [39]. Therefore, in the marketing literature, the concept of GDP per capita is understood as the sum of the total gross wealth produced in the economy able to impact consumer spending [40]. In the smartphone purchase setting, Jamalova1 and Milán [21] define the concept of GDP per capita as a financial factor enabling consumers to evaluate alternative offers. The concept of GDP per capita thus may influence the choice and expense forms of consumers in the economy. In this current study, SCP refers to smartphone choice preferences according to the available alternative, and PUR refers to the number of smartphones purchased during the transaction by the customer.

Price refers to a product's value in terms of the money a customer must spend to own the product [47]. According to Hride et al. [48], the price is one of the product's fundamental values and positively influences customer purchase and decision-making. For that reason, buyers refer to that factor in evaluating alternative offers. Considering price in product evaluation generally contributes to making a good choice [48]. Therefore, consumers tend to be more satisfied when their economic situation matches the product price [49-50-51]. Lim et al. [51] pointed out that since consumers may be sensitive to price due to their purchasing capability, focusing on factors that influence price perception is imperative because it allows them to evaluate alternatives and significantly affects customers' purchase and choice preferences.

A smartphone's ROM and RAM are described as quality and specification characteristics that the consumer relies on to assess the merit of the smartphone before making their choice [52]. In this perspective, Bhattacharjee et al. [53] consider ROM and RAM as building blocks of SCP and purchase intention. Pasha and Mu [52] pointed out that Read Only Memory (ROM) and Random Access Memory (RAM) should be considered as a criterion of consumer decision-making. Supporting that view, Bhattacharjee et al.[53] showed that these two features impact SCP and purchase. Therefore, this study considers ROM and RAM features as consumer choice and purchase criteria.

Reviewer #2: - the source of access to the data used in the research, i.e. the chosen Cross-Border E-Commerce platform, was indicated too enigmatically; please specify what data and how they were made available, which are publicly available, how they are collected, since according to the authors they are a reliable source.

Thanks for your helpful comments in revising the manuscript. For convenience, we tried our best to clarify the data source. We answered your questions about how they were made available, which part of the data is publicly available, and how the data was collected. We have explained it as follows:

The study's sample comprises four smartphone product brands (i.e., Huawei, Xiaomi, Oppo, and Samsung) purchased through the B2C selling mode by international online buyers on a Chinese CBEC platform. These buyers come from 101 countries worldwide. 

The data used mainly comes from two different sources. i.e. A private online data store (http://www.100ec.cn/Index/dsb_product.html) and a public source which is the Aliexpress platform. About 1/4 of the data comes from the private online data store, which added 3/4 of the data we downloaded directly from the Aliexpress platform using Octopus Software.

However, it should be noted that the data we got from that private source also came from the Aliexpress platform. The study's data, thus, are made of smartphone purchases from the Aliexpress platform and shipped to consumers outside China. Approximately 19,400 transactions corresponding to 24,043 smartphones purchased were obtained after combining data from the two sources mentioned above. Thus, data collection, cleaning and reorganization were done from June 30, 2022, to July 25, 2022. The whole data can be seen from this link: (https://docs.google.com/spreadsheets/d/1XeqEhDoAgm4r7Mm5yWkF9D31enwGC_4HMtIT4FVYc5Y/edit#gid=1691249773) 

AliExpress is one of the leading platforms for B2C sales. As an internationally leading platform for B2C sales, the AliExpress platform sells to about 150 million consumers from 190 countries worldwide, with more than 2400 billion visitors every year and over 100 million products [58]

---

## [Editor Report · Decision Letter 2]

26 Apr 2023

What Matters for International consumers’ Choice preferences for Smartphones: Evidence from a Cross-border Ecommerce Platform

PONE-D-22-29643R2

Dear Dr. Karamoko N'da,

We’re pleased to inform you that your manuscript has been judged scientifically suitable for publication and will be formally accepted for publication once it meets all outstanding technical requirements.

Kind regards,

Vincenzo Basile, PhD

Academic Editor

PLOS ONE

Additional Editor Comments:

Please provide a final paper with all revisions made and I recommend an additional check on plagiarism and/or compliance with the Journal's guidelines.
---

## [Editor Report · Acceptance letter]

2 May 2023

PONE-D-22-29643R2 

What Matters for International consumers’ Choice preferences for Smartphones: Evidence from a Cross-border Ecommerce Platform 

Dear Dr. N'da:

I'm pleased to inform you that your manuscript has been deemed suitable for publication in PLOS ONE. Congratulations! Your manuscript is now with our production department. 

Kind regards, 

on behalf of

Dr. Vincenzo Basile 

Academic Editor

PLOS ONE